# The Impact of Class III Obesity on Outcomes for Vestibular Schwannoma Surgery: A Case Report

**DOI:** 10.3390/diagnostics15070888

**Published:** 2025-04-01

**Authors:** Tomaž Šmigoc, Hojka Rowbottom, Janez Ravnik

**Affiliations:** Department of Neurosurgery, University Medical Centre Maribor, 2000 Maribor, Slovenia; tomaz.smigoc@ukc-mb.si (T.Š.); hojka.rowbottom@ukc-mb.si (H.R.)

**Keywords:** vestibular schwannoma, severe obesity, postoperative complications, CSF leak

## Abstract

**Background and Clinical Significance**: Vestibular schwannomas (VS) are benign tumors arising from Schwann cells of the eighth cranial nerve. They represent approximately 8% of all intracranial tumors and have an increasing incidence. Larger VS can cause brainstem compression and hydrocephalus, and magnetic resonance imaging (MRI) is the diagnostic modality of choice. Individuals with VS and an elevated body mass index (BMI) can have more postoperative complications due to their weight, which can also negatively impact the preoperative diagnostic process and planning, as well as the surgery itself, as compromises must be made since optimal positioning of the patient is often not feasible. Increased BMI is a recognized risk factor for cerebrospinal fluid (CSF) leak after microscopic resection of a VS. **Case Presentation**: This report presents a case of a patient with class III obesity who had to undergo a right VS resection with preexisting hydrocephalus and the obstacles encountered by the surgical team throughout the diagnostics process since MRI could not be performed and preoperative planning had to be based on computed tomography (CT) scan; operative treatment, where suboptimal patient placement was achieved for a planned retrosigmoid approach to the pontocerebellar angle (PCA) and postoperative rehabilitation, which was hindered by his high BMI (55 kg/m^2^) with several complications, such as CSF leak, due to his extreme weight. **Conclusions**: Despite barriers, optimal tumor resection was obtained with a long neurorehabilitation process, with a favorable outcome, emphasizing the role of a multidisciplinary team.

## 1. Introduction

Schwannomas represent benign tumors consisting of a clonal population of Schwann cells that have undergone cystic and degenerative changes and are typically attached to peripheral nerves, with most cases being sporadic [1,2]. Vestibular schwannomas (VSs) are tumors arising from Schwann cells of the eighth cranial nerve, which lies next to the seventh cranial nerve, with both nerves leaving the brainstem just below the pons and entering the internal auditory canal with the eighth cranial nerve being responsible for hearing and balance and VSs represent around 8% of all intracranial tumors [3,4,5,6,7,8,9]. The majority of VSs are in the posterior fossa, which extends from tentorium cerebelli superiorly to foramen magnum inferiorly, specifically in the angle between the pons and the cerebellar hemisphere (pontocerebellar angle) [10]. Epidemiological research conducted thus far has demonstrated an increase in the incidence [4,9]. VS are equally common in men and women [11]. The median age of VS presentation is 50 years, and it is unilateral in more than 90% of patients [7]. Clinical symptoms of VS vary, including asymmetrical hearing loss, tinnitus, vertigo, ataxia, loss of balance, and headaches [5,6]. Larger tumors can cause hydrocephalus and brainstem compression [12]. The diagnostic modality of choice is magnetic resonance imagining (MRI) with contrast-enhanced T1-weighted scans considered to be the gold standard for the initial evaluation and postoperative assessment of recurrence or residual tumors with computed tomography (CT) playing a complementary role in the evaluation of VS [13]. An audiogram, which in cases of VS shows an asymmetric sensorineural hearing loss, should also be performed in the diagnostic process [14,15,16]. Treatment options include surgery, observation with serial imaging, and stereotactic radiosurgery [6].

The population of overweight and obese individuals has reached epidemic levels and continues to rise [17]. Body mass index (BMI), based on an individual’s height and weight, has been used as a simple tool to classify people into different groups with those whose BMI ranges between 25 kg/m^2^ and 29.9 kg/m^2^ being classified as overweight and those with BMI 30 kg/m^2^ or over as obese, which can be subdivided into class I obesity (30 to less than 35 kg/m^2^), class II obesity (35 to less than 40 kg/m^2^), and class III obesity or severe obesity (40 kg/m^2^ and more) [18,19]. Elevated BMI negatively impacts postoperative outcomes in several surgical specialties, including neurosurgery [20]. Obesity can also negatively impact the diagnostic process, which cannot be performed optimally as diagnostic machines often have maximal weight allowance; hence, adjustments and compromises must be made [21]. Outcomes of a neurological examination can also be impaired by obesity, especially when assessing coordination and gait disturbances [22]. Increased BMI has been recognized as a risk factor for postoperative cerebrospinal fluid (CSF) leak, as well as longer rehabilitation and recovery, with patients often being readmitted or even discharged to a facility other than their home [20,23].

In this case report, we present our unique experience with pontocerebellar schwannoma resection in a patient with class III obesity with the immense obstacles encountered throughout the whole process and compromises we had to make as well as postoperative complications. Additionally, with the literature review, we attempted to compare our experience with the existing body of knowledge and emphasize the issues of severe obesity in neurosurgery, especially when dealing with posterior fossa lesions.

## 2. Case Report

### 2.1. History and Diagnostic Process

A 48-year-old man presented to the neurosurgical outpatient clinic at the University Hospital in Maribor, Slovenia, with hearing loss in his right ear that had been worsening for 4 years. An ENT consultant had previously examined him and found signs of sensorineural hearing loss on the right side. The neurological examination revealed hearing loss in the right ear, and on the Romberg test, he swayed to the right but did not fall. The patient also presented with gait ataxia, limiting his daily activities, and mild signs of intracranial hypertension. He suffered from diabetes mellitus type 2, arterial hypertension, dyslipidemia, and obstructive sleep apnea. Before the patient came to the neurosurgical outpatient clinic, a CT scan of his head (Figure 1) was performed, which showed a 4 × 3 cm tumor in the right cerebellopontine angle, Koos grade IV, which was pushing the right cerebellar hemisphere and the fourth ventricle to one side, closing the Luschka foramen and causing mild obstructive hydrocephalus. As the patient was claustrophobic and his BMI was 55 kg/m^2^, MRI could not be performed.

In cases of claustrophobia, an MRI can be performed under general anesthesia; however, this was not an option with our patient. Due to the weight limit of the MRI patient table and the size of the bore, which was too small for the patient’s upper body, a suboptimal diagnostic modality, a CT scan, had to be utilized for presurgical planning. Based on the lesion’s location and relative isolation from the surrounding tissues, as well as the patient’s neurologic status, a VS was suspected; however, our surgical team was extremely limited with information about the relations between the tumor and surrounding structures, such as the brainstem, cerebellar hemisphere, arteries, veins, and cranial nerves.

### 2.2. Surgery

The patient was presented at a multidisciplinary meeting where different treatment options were discussed. Due to the tumor size with an already existing hydrocephalus and the patient’s age, surgery was suggested, and he signed the consent form after being informed of the risk of surgery and additional complications due to his class III obesity. Before the operation, the patient was thoroughly examined by the attending anesthetist, as intubation and ventilation problems were to be expected. Additionally, a chest X-ray was performed before surgery, which, due to the patient’s obesity and preexisting sleep apnea, showed signs of poor ventilation and a wider mediastinum. On the day of the operation, he was taken to the operating theatre and, despite the patient exceeding the recommended weight limitations, was placed on the regular operating table. The first obstacle was mask ventilation before induction of general anesthesia, which required two people: one to hold the mask and the other to ventilate manually. During orotracheal intubation, the anesthetist had to stand on a set of steps and use a video laryngoscope at the same time. Once the airway was secured, the surgical team, with the help of the anesthesia team, began to place the patient in a “park bench” position, as a tumor resection was planned through a retrosigmoid approach. Due to the extreme weight, he could not be properly positioned on his left side but was simply tilted onto his left side, strapped down with several Velcro straps above his hips, knees, and legs. The head, which was in the Mayfield skull clamp, was rotated 90 degrees to the left to gain access. Correct positioning of the head for the desired retrosigmoid approach was impossible due to the weight of the head, a short neck, and excessive adipose tissue. As the “park bench” position was not achieved, the approach was obstructed by the patient’s shoulder and chest. Additionally, a surgical navigation system could not be used. Before the operation began, the electrodes for intraoperative neuromonitoring of the facial nerve were inserted into the right side of the face, but the signal was weak due to a thick layer of subcutaneous fat tissue. After positioning the patient, a retrosigmoid approach was attempted to remove the tumor. During surgery, part of the right cerebellar hemisphere had to be removed; otherwise, visualization of the cerebellopontine angle (CPA), where the tumor was located, would have been impossible. Due to the suboptimal patient positioning owing to his extreme weight, the surgical trajectory was incorrect, and despite basal cisternostomy and edema reduction, there was still not enough space to visualize the tumor in the CPA; hence, part of the right cerebellar hemisphere had to be sacrificed. A surgical ultrasound aspiration device was used to remove the lesion. During surgery, the posterior inferior cerebellar artery and cranial nerves V to XI were visualized, and it transpired that the tumor was firmly attached to the brainstem. Intraoperative neuromonitoring for the facial nerve was performed throughout the time of tumor removal, but the signal was extremely weak and, therefore, unreliable as the electrodes were not long enough to ensure adequate monitoring. When most of the tumor was removed, a dural substitute was used for closure, and the previously removed bone was not returned as the flap was small and for fear of brain edema. Due to the patient’s weight, the preoperative preparation took 80 min, and the surgery took 9 hours and 40 min. Straight after surgery, a CT scan was performed, which showed a postoperative condition with a removed tumor and without a significant hematoma. The patient was transferred to the intensive care unit while still on mechanical ventilation. On the first postoperative day, the patient was extubated. His cardiorespiratory condition was stable, but there was peripheral facial nerve palsy on the right side of his face. On the second postoperative day, he was transferred back to the Department of Neurosurgery, where early postoperative neurorehabilitation was started, but it was extremely difficult due to his weight and right limb ataxia. After one week, he was able to walk with a roller walker. Due to the paralysis of the peripheral facial nerve, a moist chamber was used at night, and artificial tears were administered regularly during the day. Despite great efforts by the physiotherapist, the patient was unable to walk independently. Due to the swelling of the right lower extremity, an ultrasound scan was performed to rule out deep vein thrombosis or thrombophlebitis; low-molecular-weight heparin was administered prophylactically. The dressing of the surgical wound was changed regularly, and there were no signs of inflammation or cerebrospinal fluid leakage. 11 days after surgery, the patient was transferred to a regional hospital for further rehabilitation. The pathohistological report stated that the removed tumor was a VS, a benign tumor, and, therefore, no oncological treatment was required, but complex rehabilitation was planned.

### 2.3. Postoperative Follow-Up and Complications

At the regional hospital, physiotherapy and rehabilitation continued as planned, and the patient was mobile with the aid of a roller walker. After a week in the regional hospital, the patient developed severe headaches and confusion. A CT scan was performed, which showed postoperative changes but no dynamics compared to the previous scan after surgery. The blood tests showed high inflammatory markers, and a urinary tract infection was also diagnosed. As a central nervous system infection was suspected, a course of antibiotics (metronidazole, ampicillin, ceftriaxone) was prescribed, which was administered for 6 days and exchanged for flucloxacillin when the inflammatory markers decreased. A lumbar puncture was also performed, but the microbiological tests showed no microorganisms in the cerebrospinal fluid. Approximately one month after surgery, a CSF leak occurred through the surgical wound, and the patient was transferred back to the tertiary center. A further CT scan showed a large soft tissue edema around the craniectomy. Flucloxacillin was replaced by vancomycin and meropenem. During his hospitalization at the Department of Neurosurgery, a lumbar drain was inserted to control the leakage of cerebrospinal fluid through the surgical wound. The placement of the drainage was challenging since the needle was barely long enough to reach the spinal canal. During his 20-day readmission to the tertiary center, the CSF leakage stopped, and it did not reoccur after the removal of the lumbar drainage. The wound healed completely, the inflammatory markers were low, and he was discharged home, equipped with a roller walker and a wheelchair, both of which he learned to use unaided. On discharge, he was prescribed linezolid for 14 days as an extension of the previous intravenous antibiotic treatment. As planned during his hospitalization, he underwent complex rehabilitation in a tertiary healthcare facility and slowly regained his independence in daily activities. Four months after his discharge from the neurosurgery department, he returned to the outpatient clinic. The follow-up CT scan showed postoperative changes with no tumor residue or recurrence. He was able to walk at home without the roller, but the peripheral facial nerve palsy persisted, and he was regularly examined by an ophthalmologist and a plastic surgeon. He continued the use of artificial tears in the day and the moister chamber at night. Two years after the initial surgery, the control CT scan (Figure 2) showed no signs of residual tumor or recurrence. At his last appointment in the neurosurgical outpatient clinic, the patient was mobile with no aid, and due to a persistent facial nerve palsy to the right side, facial reanimation surgery was performed to restore symmetry.

## 3. Discussion

### 3.1. Vestibular Schwannomas

The first successful VS surgery was performed in 1894 with an initial mortality of almost 80%, however, with the development of the operating microscope, refinement of surgical techniques as well as improvements in antibiotics, the current mortality is approximately 0.2%. Despite advances in surgery, the number of patients treated surgically has been slowly declining, especially for smaller tumors, with an increase in patients being observed with serial imaging or treated with stereotactic radiosurgery [6].

Observation is an option for smaller tumors, older patients, and those with major comorbidities. MR imagining is the preferred technique that provides incomparable tumor characteristics, whereas contrast-enhanced CT of the temporal bones can be utilized as an alternative when the patient cannot undergo MR imagining [7]. In obese individuals with VS, stereotactic radiosurgery might be a superior treatment modality, depending on the size of the lesion and degree of hearing loss as well as the presence of other symptoms [5]; however, that was not an option in our case due to tumor’s size, mass effect, and existing hydrocephalus. Additionally, targeted biological therapies, such as bevacizumab, are currently emerging as options for the treatment of VS; however, at the moment, that is only applicable to patients with neurofibromatosis type 2 that have a VS [12].

Surgical excision remains the definitive treatment for larger lesions and younger individuals [6]. VS may be approached by a translabyrinthine, retrosigmoid, or middle fossa craniotomy [7]. Serious complications after surgery, such as stroke, CSF leak, wound infection, intracranial bleeding, and meningitis, are infrequent [6]. Nowadays, the aim of surgery has moved from total resection to functional preservation, especially in cases where the entire tumor mass cannot be safely removed concerning cranial nerves [7].

Being a benign tumor with a slow-growing nature, VS allows a postponement of surgical treatment to mitigate lifestyle factors, such as obesity, before deciding to operate [5]. A perioperative loss of weight is associated with a reduction in deep surgical site infections, abscess formation, and minor wound complications [20]. Weight reduction was proposed to our patient; however, cooperation was not great. Additionally, limited by his diabetes and due to evolving hydrocephalus, there was not enough time for him to lose the required weight, and surgery had to be performed despite the increased risk of complications.

### 3.2. Obesity in Neurosurgery

The population of overweight and obese people is rising, and in places like the United States of America as well as Europe, it has reached epidemic levels with an estimated prevalence of 42% [17] and has increased by around 50% per decade over the past 20 years. This has been seen among both sexes, all age groups, and races [23,24]. In Slovenia in 2020, 39% of inhabitants were overweight, and 20% were obese, with males and poorly educated individuals being the most affected. The patient presented in this case also comes from a region of Slovenia that has the highest percentage of obese individuals (25.7%), which could be due to cultural differences and mainly rural households with predominantly meat-based diets, processed meat products, and lard used in cooking [25].

Elevated BMI negatively impacts postoperative outcomes in several surgical specialties, including neurosurgery, and it is proven that a BMI above 40 kg/m^2^ represents an independent risk factor for perioperative complications [20,26]; these individuals have a higher incidence of obstructive sleep apnea, idiopathic hypertension and even spontaneous CSF leaks [17]. Individuals with class III obesity (BMI above 40 kg/m^2^) have a higher risk of surgical site infections, which can be a consequence of longer operating time with greater risk of wound contamination as well as prolonged retraction-related ischemia. Additionally, chronic inflammation and dysfunction of the immune system that is present in obesity and metabolic syndrome can increase the chances of surgical site infection [20,27].

CSF leak remains a common postoperative complication following the microscopic resection of a VS [5,28], which leads to extended hospitalization as well as higher patient morbidity and mortality. Elevated BMI has been recognized as a modifiable risk factor for surgical site infections [28]. Additionally, there is a 22% chance of developing a CSF leak in individuals with class III obesity [20]. Postoperative CSF leak increases the risk of meningitis and reoperation and prolongs hospital stay [17].

It has been proven that obese individuals have elevated intraabdominal pressure, which leads to a higher intrathoracic and cardiac filling pressure, thereby leading to an increase in the baseline intracranial pressure (ICP), which can lead to a postoperative CSF leak [28]. An increase in the baseline ICP could also be due to the fat pannus and its negative effects on venous return, increased levels of serum estrogens, and hypercapnia from obstructive sleep apnea that leads to cerebral vasodilatation, hence further increasing the ICP [20,28].

Additionally, obese patients are likely to experience longer operative times since obesity is linked with a difficult airway, which can prolong the induction of anesthesia. Also, positioning for the surgical procedure can present numerous challenges, thereby prolonging surgical duration [17,29]. Intubation of obese patients is challenging due to the excessive facial, palatal, and pharyngeal soft tissue and impaired jaw, atlantoaxial, and cervical mobility. Pharmacokinetics and pharmacodynamics of medications are altered in obese individuals. Increased positive end-expiratory pressure, a higher fraction of inspired oxygen, and inotropes are often required in obese individuals [20]. An obese patient requires a greater diaphragmatic excursion to provide the same degree of ventilation, their functional residual capacity, expiratory reserve volume and total lung capacity all being decreased [30].

Proper positioning of the patient is vital for a successful neurosurgical procedure, both for a trajectory to the lesion as well as to avoid complications to non-operated-upon tissues. The park bench position, which allows access to the lesion of the cerebellopontine angle, represents a modification of the lateral position with at least one arm positioned outside of the operating table, providing a greater manipulation of the head and neck [30]. A lumbar drain, placed at the time of surgery, can aid and allow better access to a posterior fossa in cases of elevated baseline ICP, and if it remains in place postoperatively for a short period, it can decrease the risk of developing a CSF leak [28].

The recovery time in obese patients tends to be longer [20]. Also, in park bench position, axillary misplacement can lead to limb ischemia, whereas excessive flexion of the torso can lead to brachial plexus injuries when using rigid head fixation. Obese patients are at an increased risk of ulceration during surgery as well as position-related peripheral nerve injury, especially ulnar and lateral femoral cutaneous nerve [30].

### 3.3. Our Experience with Severe Obesity and VS Surgery

Due to his class III obesity, the patient in our case report could not undergo MR imaging, which would have been a preferable and superior technique to the CT scan performed. As the diagnostic imaging revealed obstructive hydrocephalus, he had to be treated despite his obesity and anticipated complications, as there was no time for preoperative weight loss. Due to his age and the size of the neoplasm, surgical treatment was proposed. His obesity was a major problem for both the anesthetic and surgical teams throughout the operation. The preoperative preparation time was prolonged to 80 min, and several issues with ventilation were encountered by the anesthetists, with two consultants present in case of airway or breathing issues. The positioning was suboptimal, as he was only tilted because his extreme weight prevented him from being placed in a correct park bench position, which is required for a retrosigmoid approach. As the position of the body and the head were substandard, the resection was hindered by poor visibility and a lack of maneuvering space for surgical instruments, which was also affected by the position of the patient’s shoulder and chest. During surgery, longer instruments had to be utilized. When comparing our surgical time with other cases of CPA VS resection, it took us almost twice as long to resect it. In addition, the electrodes used for intraoperative neuromonitoring of the facial nerve were not long enough due to a thick layer of subcutaneous adipose tissue, so neuromonitoring was unreliable and could not be used as an aid during surgery, which further complicated the resection, as we were also unable to utilize the navigation system.

His class III obesity was also a major obstacle to his postoperative neurorehabilitation, as he required four physiotherapists to assist him with a rollator to prevent him from falling. The nursing team also had issues with taking daily care of the patient due to his severe obesity, as the department did not have specific hoists to help move the patient; therefore, several nurses had to attend to him. In cases of poorly mobile obese individuals, surgical wounds can be subject to less-than-optimal medical care, dehiscence, and consecutive surgical site infection. A cerebrospinal fluid leak, which occurred due to a combination of substandard conditions during resection surgery, increased intracranial and intraabdominal pressure, diabetes, wound dehiscence, and infection, was successfully treated conservatively with antibiotics and the insertion of a lumbar drain. However, the condition significantly prolonged his hospitalization. Had we not been successful with the lumbar drainage, the team planned to revise surgery with another attempt at duroplasty or to try and insert ventriculoperitoneal drainage, which would have been challenging in his case and associated with additional risks and complications, such as migration of the peritoneal catheter and infections.

Obese patients, after surgical treatment, are readmitted more often than nonobese individuals [20,23]. Obese patients are more often discharged to a facility other than their home compared to their normal-weight counterparts [20]. The same was true in our case since the patient had to be hospitalized for 14 days after the first surgery, and he also required additional rehospitalization at our tertiary institute as well as prolonged hospitalization at his community hospital. His recovery was lengthy, with early rehabilitation orientated towards basic everyday activities, which took him 6 months to accomplish and another year to become unaided when walking and further surgery for his right-sided peripheral facial nerve palsy to be corrected.

## 4. Conclusions

Obesity represents an important risk factor for brain tumor resection surgery as it can mask clinical symptoms and signs typical of brain neoplasms. Due to severe obesity, preoperative diagnostic work-up is often limited. In cases of benign brain lesions, it is vital to preoperatively provide patients with good locomotor and respiratory physiotherapy, an appropriate nutritional regime, and to provide good control over their comorbidities. It is vital that preoperatively and postoperatively anesthetists are prepared for possible complications with intubation and timely extubation. Surgical preoperative preparation in cases of severely obese patients consists of adjustments to the placement of the patient, careful preoperative planning of the resection, and application of additional safety nets, such as intraoperative neuromonitoring; however, the surgical team must be prepared for sudden and unexpected rearrangements. The patient must be made aware of the increased risk of surgery due to their obesity. When a patient is severely obese, the operating times are longer, postoperative neurorehabilitation is longer, and those patients tend to be hospitalized more often than their counterparts with normal BMI. When considering the risk factors, required adaptations, good teamwork, and patient cooperation, the result of resection of benign brain lesions in severely obese individuals can have good results.

## Figures and Tables

**Figure 1 diagnostics-15-00888-f001:**
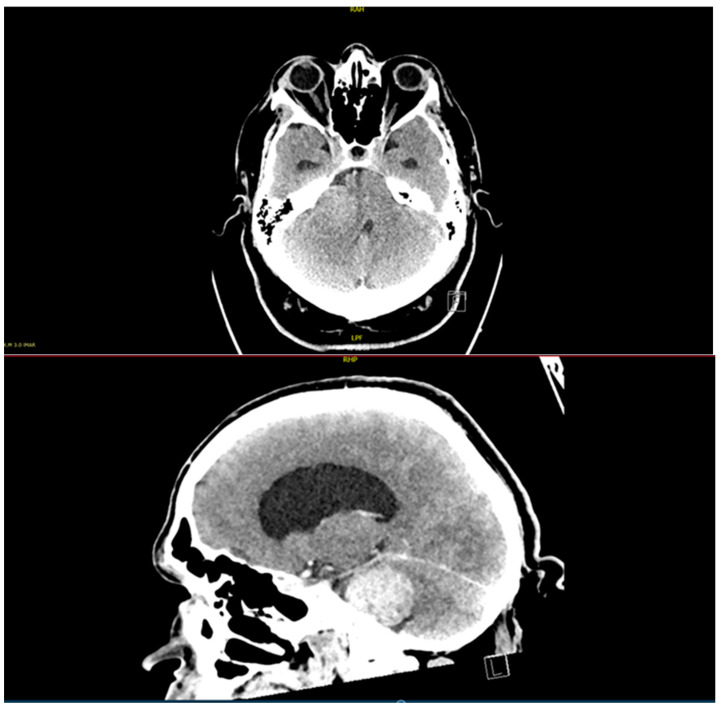
Preoperative CT scan (axial and sagittal view after contrast enhancement) showing a lesion (VS) in the right CPA measuring 4 × 3 cm with obstructive hydrocephalus due to compression of the 4th ventricle.

**Figure 2 diagnostics-15-00888-f002:**
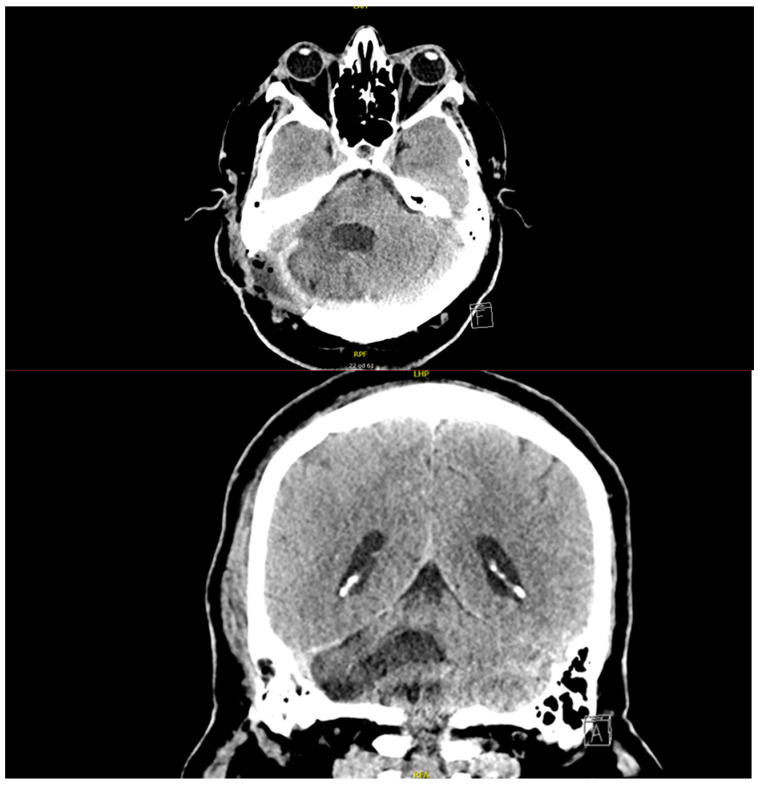
CT scan performed 2 years following the initial surgery, showing no signs of residual tumor or reoccurrence with encephalomalacic changes in the right cerebellar hemisphere. The hypodense subcutaneous inclusions represent the residue of bone wax and fatty tissue used to close the mastoid air cells to prevent rhinorrhea/otorrhea.

## Data Availability

The raw data supporting the conclusions of this article will be made available by the authors upon request.

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
