# Peer review of "The Impact of Class III Obesity on Outcomes for Vestibular Schwannoma Surgery: A Case Report"

_diagnostics, 2025, doi:10.3390/diagnostics15070888_

Round 1
Reviewer 1 Report
Comments and Suggestions for Authors
The authors present an interesting and challenging case report of a 48-years-old male patient, suffering of obesity class III, having other comorbidities (DM 2, arterial hypertension, dyslipidemia, and obstructive sleep apnea). He received a diagnosis of VS on the right side, associated with worsening hearing loss. His BMI and claustrophoby did not allow to get a pre-operative MRI but only a CT scan was performed, then retrosigmoid approach was performed for surgery, CSF leakage was the main postoperrative complication. The rehabilitation process was longer and complicated, finally achieving a favourable outcome.
1) Can please the authors specify more details about the lesion, according to the KOOS grading scale? Minor: I suggest to precise the side of the lesion even in the abstract
2) Treatment options include surgery, observation with clinical and radiological follow-up, or stereotactic radiosurgery: can the authors explain why they did not choose GammaKnife instead of open surgery ( considering patient's characteristics and comorbidities)? Did they discuss this option with their patient? was there any preoperative multisciplinary tumour board meeting to decide what was the best treatment option?
3) the patient presented with preexisting hydrocephalus, did the authors consider to perform first a thrid ventriculostomy to avoid obstructive hydrocephalus and allow a better exposure of the surgical corridor,usign a retrosigmoid approach later on?
Many thanks for sharing your experience on a fascinating but challenging surgery
Author Response
The authors present an interesting and challenging case report of a 48-years-old male patient, suffering of obesity class III, having other comorbidities (DM 2, arterial hypertension, dyslipidemia, and obstructive sleep apnea). He received a diagnosis of VS on the right side, associated with worsening hearing loss. His BMI and claustrophoby did not allow to get a pre-operative MRI but only a CT scan was performed, then retrosigmoid approach was performed for surgery, CSF leakage was the main postoperrative complication. The rehabilitation process was longer and complicated, finally achieving a favourable outcome.
- Can please the authors specify more details about the lesion, according to the KOOS grading scale? Minor: I suggest to precise the side of the lesion even in the abstract
Response: Thank you for your comment. In the section History and diagnostic process, we added that the schwannoma was a Koos grade IV, and in the Abstract we added that the schwannoma was located on the right side as proposed.
- Treatment options include surgery, observation with clinical and radiological follow-up, or stereotactic radiosurgery: can the authors explain why they did not choose GammaKnife instead of open surgery ( considering patient's characteristics and comorbidities)? Did they discuss this option with their patient? was there any preoperative multisciplinary tumour board meeting to decide what was the best treatment option?
Response: Thank you for your comment. The patient was presented at a multidisciplinary board meeting, where treatment options were analysed and based on the presence of an obstructive hydrocephalus with pressure on the brainstem, we opted for surgery, despite patient's weight as our main aim was to remove the tumor, alleviate the pressue on the brainstem and allow for a normal cerebrospinal fluid flow. We also discussed different treatment option, their pros and cons with the patient prior to surgery.
Due to the size of the schwannoma in this case and an already existing hydrocephalus, we did not opt for gamma knife surgery or other type of stereotactic surgery, since in cases of Koos grade III and IV, microsurgical resection achieves better long-term tumour control, and with the existing hydrocephalus we were affraid of peritumoral edema developing postradiation, thus, worsening the neurological status.
- The patient presented with preexisting hydrocephalus, did the authors consider to perform first a thrid ventriculostomy to avoid obstructive hydrocephalus and allow a better exposure of the surgical corridor,usign a retrosigmoid approach later on?
Response: Thank you for your comment. Before surgery, we considered solving hydrocephalus first, however, we wanted to achieve as much as possible in one surgery as his obesity did no present a challenge for the surgical team, but it was also extremely demanding for the anaesthetist as the intubation and ventilation posed a problem and after surgery, the patient had to be transfered to the intensive care to be safely extubated. One could argue, that we could have done a third ventriculostomy and schwannoma resection in one surgery, however, that would then even further prolonge the lenght of our surgery, which was already extended.
Before attempting to resect the schwannoma, we performed a basal cisternostomy from cisterna magna. However, despite that being successful, there was still not enough space to optimally visualise the cerebellopontine angle since the whole approach and positioning were suboptimal, and we believe that even with a 3rd ventriculostomy, we would encounter similar issues. We also did not opt for a ventriculoperitonealn shunt in his case based on his severe obesity and possible further complications.
Reviewer 2 Report
Comments and Suggestions for Authors
This paper reports on the removal of a large vestibular schwannoma in an obese patient with multiple comorboidities. Unfortunately, there is not much elese to say, because this is surely not the only case of a severely obese patient operated for such problem and I would say the authors took a great risk with the operation. What clearly emerges is that everything in this type of patients is suboptimal and that, to the aim of removing a tumor which was only causing hearing loss, they had to remove parto of the cerebellar hemisphere, because of the technical challenge represented by the inadequate position and the difficulties in trajectory, together with the impossibility of using neuronavigation. The only question that could be asked if it was worthy, because of the facial nerve palsy and the cebellar damage, that in a barely mobile patient could be a cause forlong term immobilization and death, something that did not happen. So, beyonbd commending the authors for their life saving procedure, I would say this paper does not add anything new. For a case rteport it is far too long too. I am sorry but I do not see any good reason to support publication
Author Response
Reviewer 2
This paper reports on the removal of a large vestibular schwannoma in an obese patient with multiple comorboidities. Unfortunately, there is not much elese to say, because this is surely not the only case of a severely obese patient operated for such problem and I would say the authors took a great risk with the operation. What clearly emerges is that everything in this type of patients is suboptimal and that, to the aim of removing a tumor which was only causing hearing loss, they had to remove parto of the cerebellar hemisphere, because of the technical challenge represented by the inadequate position and the difficulties in trajectory, together with the impossibility of using neuronavigation. The only question that could be asked if it was worthy, because of the facial nerve palsy and the cebellar damage, that in a barely mobile patient could be a cause forlong term immobilization and death, something that did not happen. So, beyonbd commending the authors for their life saving procedure, I would say this paper does not add anything new. For a case rteport it is far too long too. I am sorry but I do not see any good reason to support publication
Response: Thank you for your comments. When the patient presented to the neurosurgical outpatient clinic, he complained of hearing loss. Still, upon further examination, we discovered gait ataxia limiting his daily activities and signs of intracranial hypertension as he already had an obstructive hydrocephalus from the vestibular schwannoma. In the first paragraph of the section History and diagnostic process, we added additional information about the patient's neurological status before surgery to shed light on our decision for surgery. Despite the surgery being challenging and his postoperative rehabilitation long, the patient's mobility post-surgery was better than before as gait ataxia subsided. He had also been examined by a plastic surgeon who performed a facial reanimation surgery to restore the symmetry of his face.
The authors compose this case report since we do not have an abundance of experience with severely obese patients with vestibular schwannomas and we found it challenging throughout the whole process and wanted to share our experience, and also to further emphasise the potential negative effects of severe obesity on neurosurgical patients. Of course, ours is not the first nor the only report of that sort, however, with the increasing numbers of overweight and obese individuals, it is important to raise awareness of not just postoperative complications linked with obesity, namely surgical site infections or cerebrospinal fluid leak, but also problems encountered in the diagnostics process, during intubation, during positioning and throughout surgery and even more in the rehabilitation process. We also performed a literature review to compare our work with the works of others in the field and we believe that it is an important topic that should be discussed further.
Reviewer 3 Report
Comments and Suggestions for Authors
A very good case report about the complicated surgery in obese patient. The case description is adequate. I have the questions about the illustrative postoperative CT images (Figure 2) two years after the surgery: there are hipodense subcutaneous inclusions on axial slice - are there air bubbles? bone wax? Please add the comment or explanation for this. The second remark is about the very explicit statement in discussion section (lines 354-357), that "...so neuromonitoring was unreliable, resulting in postoperative peripheral facial nerve palsy". This complication is usually produced by the surgeon and the combination of unfavorable circumstances. I suggest to change the form of a statement to a more delicate one.
Author Response
Reviewer 3
A very good case report about the complicated surgery in obese patient. The case description is adequate. I have the questions about the illustrative postoperative CT images (Figure 2) two years after the surgery: there are hipodense subcutaneous inclusions on axial slice - are there air bubbles? bone wax? Please add the comment or explanation for this.
Response: Thank you for your comments. The hypodense subcutaneous inclusions are the residue of bone wax and fatty tissue used to close the mastroid air cells to prevent rhinorrhea/otorrhea.
The second remark is about the very explicit statement in discussion section (lines 354-357), that "...so neuromonitoring was unreliable, resulting in postoperative peripheral facial nerve palsy". This complication is usually produced by the surgeon and the combination of unfavorable circumstances. I suggest to change the form of a statement to a more delicate one.
Response: Thank you for your comments. We agree that the sentence could be potentially misinterpreted, hence, we removed the second part of the sentence and we added that as the neuromonitoring was unreliable it was of no aid to us during surgery, which further complicated the resection as we were also unable to use the neuronavigation.
Round 2
Reviewer 2 Report
Comments and Suggestions for Authors
I thank the authors for their answer, that is sincere and stress the fact that the only element suggesting a potential interest for publication in a similar case is the peculiar condition of the patient. It is necessary though that they understand that not everyone can be operated on and surgery has to be a choice of the patient not less than of the surgeon. So, the main question is, the patient had gait ataxia and hearing loss. WOuld have been possible to wait for him to loose weight, let's say a year, than to perform the surgery? I know mine seems a stretch too, but surgery should not be an act of courage by those who operate. it should rather be an act of wisdom. So, even though I understand the reasons that led to surgery, the only way for me to accept the paper would be to review it again and: 1) to shorten it to half its length; 2) to discuss what other options were considered, including the possibility for the patient to undergo palliative radiosurgery, waiting for his weigth to reduce enough to be able to face surgery more safely and the possibility of not having surgery at all.
Author Response
I thank the authors for their answer, that is sincere and stress the fact that the only element suggesting a potential interest for publication in a similar case is the peculiar condition of the patient. It is necessary though that they understand that not everyone can be operated on and surgery has to be a choice of the patient not less than of the surgeon. So, the main question is, the patient had gait ataxia and hearing loss. WOuld have been possible to wait for him to loose weight, let's say a year, than to perform the surgery? I know mine seems a stretch too, but surgery should not be an act of courage by those who operate. it should rather be an act of wisdom. So, even though I understand the reasons that led to surgery, the only way for me to accept the paper would be to review it again and: 1) to shorten it to half its length; 2) to discuss what other options were considered, including the possibility for the patient to undergo palliative radiosurgery, waiting for his weigth to reduce enough to be able to face surgery more safely and the possibility of not having surgery at all.
Response: Thank you for your comments. Prior to surgery, the patient was presented at a multidisciplinary meeting where different treatment options and modalities were discussed. Based on the size of the tumor and the already existing obstructive hydrocephalus with its signs manifesting upon clinical examination as well as compression of the brainstem seen on the CT scan, the team, together with the patient, opted for surgery and not for radiation of the lesion, which would lead to reactive postradiation edema and would further worsen patient’s neurologic condition and also, looking at the mass of the tumor, the radiation would not have been effective enough. Furthermore, we thought about ventriculoperitoneal shunt or ventriculostomy prior to resection, however, it was not just the obstruction that was causing an issue, but the mass of the lesion itself and its pressure on the brainstem, which would not be alleviated by shunting the cerebrospinal fluid. Additionally, insertion of a ventriculoperitoneal shunt would be challenging in our case, since due to severe obesity, the intraabdominal pressure would be increased, which could potentially impair the function of the shunt and it would mean exposing the patient to another procedure under general anesthesia with its risks and gambling with possible postoperative complications, such as would infection or problems with healing. Due to patient’s diabetes and other comorbidities, expected issues with general anesthesia, we, in agreement with the patient, wanted to undertake as much as possible in one procedure, since for the patient every surgery would be extremely risky and linked with further perioperative and postoperative complications, namely, wound healing and infections. Everything was discussed with the patient; the aim of surgery and the potential complications due to his extreme weight. The patient in our case wanted to be operated on, since the tumor was affecting his daily life and was preventing him from normal functionality as he was unable to work and had issues with his gait and required assistance. Due to the patient’s poor mobility, diabetes and comorbidities it was unreasonable from us to expect that he would lose weight to make surgery easier with conditions more optimal and less risky. As he had already manifested with signs of increased intracranial pressure due to an obstructive hydrocephalus, we also did not have time to wait for the patient to lose suffient weight and based on the fact that he was becoming less and less mobile we were expecting a further increase in patient’s weight.
As a team, we always aspire to provide comprehensive and patient orientated treatment, not just focusing on the surgical aspects, which was also the main aim for writing this article. We also analysed the existing literature on the topic of surgery for brain tumors in extremely obese individuals and with this article, we wanted to shed light on the importance of paying attention to obstacles encountered throughout the diagnostic process, treatment and during postoperative rehabilitation and not solely focusing on performing surgery. As it is not just a case report but also an analysis of the existing body of literature, the article is longer, however, we managed to shorten it by approximately 500 words. Not much is written specifically about the resection of brain tumors in extremely obese patients and with the incidence of obese individuals on the rise, we believe it to be a relevant topic for neurosurgeons in the future.
Round 3
Reviewer 2 Report
Comments and Suggestions for Authors
I carefully read authors' answer and I find them satisfactory. So. I do believe no other work on the paper is needed